# Neutrophils’ Extracellular Trap Mechanisms: From Physiology to Pathology

**DOI:** 10.3390/ijms232112855

**Published:** 2022-10-25

**Authors:** Janina Schoen, Maximilien Euler, Christine Schauer, Georg Schett, Martin Herrmann, Jasmin Knopf, Kursat Oguz Yaykasli

**Affiliations:** 1Department of Internal Medicine 3—Rheumatology and Immunology, Friedrich-Alexander-University Erlangen-Nürnberg (FAU) and Universitätsklinikum Erlangen, 91054 Erlangen, Germany; 2Deutsches Zentrum für Immuntherapie (DZI), Friedrich-Alexander-University Erlangen-Nürnberg and Universitätsklinikum Erlangen, 91054 Erlangen, Germany

**Keywords:** neutrophil extracellular traps, inflammation resolution, DAMP, aggNETs, occlusions, autoantigens

## Abstract

Neutrophils are an essential part of the innate immune system and the first line of defense against invading pathogens. They phagocytose, release granular contents, produce reactive oxygen species, and form neutrophil extracellular traps (NETs) to fight pathogens. With the characterization of NETs and their components, neutrophils were identified as players of the innate adaptive crosstalk. This has placed NETs at the center not only of physiological but also pathological processes. Aside from their role in pathogen uptake and clearance, NETs have been demonstrated to contribute to the resolution of inflammation by forming aggregated NETs able to degrade inflammatory mediators. On the other hand, NETs have the potential to foster severe pathological conditions. When homeostasis is disrupted, they occlude vessels and ducts, serve as sources of autoantigens and danger or damage associated molecular patterns, directly damage tissues, and exaggerate complement activity and inflammation. This review focusses on the understanding of NETs from their formation to their functions in both physiological and pathological processes.

## 1. Introduction

In humans, neutrophils account for 40 to 70% of total white blood cells under physiological conditions. They are the most prevalent circulating polymorphonuclear (PMN) leukocytes. Their main role is to eliminate pathogenic agents at the sites of infectious or non-infectious inflammation. They also play roles in a variety of physiological and pathological processes and are endowed with a plethora of surface receptors for immunological and pathological mediators [1,2]. Neutrophil extracellular traps (NETs) were added to the arsenal of neutrophils in 2004 [3]. NETs are mesh-like structures made up of extruded decondensed DNA, modified histones, and more than 30 primary and secondary granular proteins. The activation of reactive oxygen species (ROS) and increase in intracellular calcium drive NET formation. The decondensed, extruded DNA is much more extended than the cell it is derived from and immobilizes and kills invaders while also acting as a physical barrier to inflammatory mediators [4,5,6]. NETs form not only in response to invaders but also in response to endogenous inflammatory signals [7,8]. While NETs beneficially eliminate invaders, neutrophils release a variety of pro-inflammatory mediators during their formation. Usually single NETs get rapidly cleared by co-operation of DNases and phagocytes. However, large aggregated NETs often persist in the tissues and shift from release to degradation of pro-inflammatory mediators. They orchestrate the resolution of inflammation. Formation and removal of NETs are delicately balanced. A shift in favor of NET formation will foster recruitment and activation of further neutrophils. This initiates a vicious circle that exacerbates and chronifies the inflammatory response. NET-driven inflammation is involved in several autoimmune diseases [7,9]. Indeed, if inflammatory NET-related signals persist for an extended period, aggregated NETs (aggNETs) can cause vascular and ductal occlusions [8]. The aim of this review is to present a general but detailed profile of NETs and their properties suitable for both scientists new to the field and for more experienced readers who wish to deepen their knowledge. We outline the role of NETs from maintenance of normal physiology to the development of pathologies. We also report on the mechanisms underlying NET formation in response to various triggers.

## 2. NET Formation Mechanisms and Their Components

DNA externalization of neutrophils was already pinpointed in 1996 as an additional type of cell death, different from apoptosis or necrosis [10]. Phorbol 12-myristate 13-acetate (PMA) caused the decondensation of chromatin, membrane rupture, and externalization of DNA–protein adducts; the process was fast and lethal. Using additional inducers, such as IL-8 and lipopolysaccharide (LPS) these data were confirmed and the term “NET” was coined [3]. The death of neutrophils occurred within several hours after induction [11]; the term “NETosis” was used to describe this process [12]. This term was widely used until the discovery of non-lethal NET formation in 2009 [13]. The discussion about the use of the term “NETosis” is ongoing. In 2018, the Nomenclature Committee on Cell Death advised not to use term “NETosis” until clear evidence of cell death is available [14]. There is still some ambiguity in the terminology, with some groups preferring to use the term “vital NETosis” [15,16]; we and others use the term “NET formation” [17,18,19]. In this review, we will use the terms “vital” and “suicidal NET formation”, respectively.

### 2.1. NADPH Oxidase (NOX)-Dependent NET Formation

A large amount of information has already been gathered on NADPH oxidase (NOX)-dependent NET formation. Several stimulants, including PMA and LPS, have been shown to trigger NET formation [11,20,21]. NOX is a multicomponent enzyme complex located in neutrophils’ plasma and phagosome membranes. It oxidizes NADPH and is responsible for the generation of superoxide anions. When neutrophils encounter certain stimulants, NOX and its subunits are phosphorylated via the Raf–MEK–ERK–PKC pathway [11,22,23]. The activation of NOX has crucial importance for NET formation and oxidative defense against invading microorganisms. The Raf–MEK–ERK pathway drives NET formation and suppresses apoptosis by activation of anti-apoptotic proteins such as Mcl-1. However, there are conflicting reports as to whether the Raf–MEK–ERK pathway and PKC act together. While PKC was found to be involved in NET formation triggered by Helicobacter pylori in conjunction with the Raf–MEK–ERK pathway, it was not involved in NET formation triggered by Entamoeba histolytica parasite [23,24]. Akt, also called protein kinase B, is another decision-maker favoring NOX-dependent NET formation over apoptosis [25]. Another NOX-dependent pathway included the activation of c-Jun N-terminal kinases (JNK) upon stimulation with LPS but not PMA [26]. The relationship between signaling pathways and types of neutrophil death is a complex issue and warrants further investigation.

Under physiological conditions, myeloperoxidase (MPO) and, among others, the homologous serine proteases neutrophil elastase (NE), cathepsin G (CatG), and Proteinase 3 form a structure referred to as an “azurosome”. ROS dissociates NE, CatG and azurocidin 1 from MPO into the cytosol. Free NE then breaks down F-actin. After nuclear membrane permeabilization, free serine proteases and MPO translocate into the nucleus. There, NE cleaves histones to facilitate the homogenization of euchromatin and heterochromatin. MPO promotes chromatin disruption and relaxation, although the exact mechanism is still elusive. The DNA decorated with granule proteins compiles in the vast intracellular vacuole and is extruded from the cell as NETs [27,28,29]. The importance of NE and MPO for NET formation is particularly evident since neutrophils that lack functional NE and MPO reportedly fail to NOX-dependently form NETs [27,28].

ROS promotes chromatin decondensation, one of the initial steps of NET formation. The levels of intracellular ROS have to be delicately balanced, as overproduction can cause serious tissue injuries and underproduction may impair the immune system. Since a report of individuals with a mutation in the NADPH oxidase gene resulting in reduced ROS production and hence NETs, it has been realized that this balance is mostly maintained through the enzyme NOX [22,30]. Another factor that affects the formation of NETs is the local pH. The optimal pH for NE and MPO activity is basic (7.5–8.5) and acidic (4.6–6.0), respectively [31]. The production of ROS and NET formation was reduced in acidic conditions, while phagocytosis and bacterial killing were not affected [31]. The rise from pH 6.6 to 7.8 augmented NOX-dependent ROS production, protease activity as indicated by higher cleavage of histone H4, and enhanced NET formation [32]. These findings highlight the importance of the CO_2_/HCO_3_– balance in the NET formation, since the alkalinization of neutrophils in response to an alkaline environment increases intracellular calcium levels and neutrophil activation [31,33].

### 2.2. NOX-Independent NET Formation

There are suicidal pathways of NET formation that do not involve NOX [34] but are driven by an increased influx of Ca^2+^ into the cells’ cytoplasm. This process is stimulated by the fungal ionophores A23187 (from Streptomyces chartreusensis), ionomycin (from Streptomyces conglobatus), and nigericin (from Streptomyces hygroscopicus). Furthermore, endogenous physiological agents such as the granulocyte–macrophage colony-stimulating factor (GM-CSF) act in a similar matter [35,36]. As the mechanism is NOX independent, the roles of NE and MPO in chromatin remodeling are still elusive.

PADI enzymes replace NE and MPO in the NOX-independent pathways. Of the five members of the human PADI family, PADI2 and PADI4 are expressed by neutrophils. PADI4 appears to be essential for NET formation [37], although a regulatory role for PADI2 has been proposed recently [38]. With a molecular weight of 74 kDa, human PADI4 contains five Ca^2+^ binding sites. Two C-terminal ones take functional roles in the conformational changes induced by Ca^2+^, while the N-terminal sites contribute to the overall stability of the PADI4 protein [39]. The extracellular Ca^2+^ that enters the cytoplasm enables the translocation of PADI4 from the cytosol into the nucleus, where it citrullinates histones. This reduces the positive charge of the histones and weakens their interaction with the negatively charged backbone of the DNA. This unique process causes chromatin decondensation. While histone H4 is reportedly citrullinated in the NOX-dependent mechanisms, in the NOX-independent pathway the massive citrullination occurs at histone H3 [40]. Both citrullinated histones H3 (citH3) and H4 (citH4) are biomarkers of NETs [6,41]. PADI4 inhibition reduces NET formation in both humans and mice [42]. Therefore, pharmaceutical inhibitors for PADI2 and PADI4 have been developed to limit excessive NET formation [37].

Another hallmark of the Ca^2+^ ionophore-induced NET formation is the production of mitochondrial ROS (mROS) by the activation of the small conductance calcium-activated potassium 3 channel (SK3) [43]. SK molecules are a subfamily of Ca^2+^-activated K^+^ channels, and their importance for immune cells, especially for neutrophils, is established [43,44]. SK3 mediates mROS production, induces apoptosis of granulocytes, and NET formation [43,44]. Mitochondria are crucial for the inflammatory response as they produce ROS and release their DNA along with nuclear DNA during NET formation. The crosstalk between mROS and NOX has already been investigated, and the role of mROS in the activation of NOX was demonstrated [45,46]. If completely ROS-independent NET formation pathways exist, they are still elusive.

Oxygen radicals trigger the cascade of the mitogen-activated protein kinases that enable membrane permeation of granules and DNA. Akt, p38, cSrc, PyK2, and JNK are engaged in the NOX-independent mechanism. Compared to the NOX-dependent pathway, activation levels of Akt and ERK are increased and decreased, respectively. Thus, ERK is essential for NOX-dependent NET formation whereas Akt orchestrates the NOX-independent pathway [6,43,47]. The NOX-independent mechanism is also affected by pH due to pH-regulated enzymes involved in NET formation. The optimal pH for PADI4 activity and NET formation is reportedly at pH 7.6–8.0 [33,48].

### 2.3. DNA Release from Viable Cells

Functions and pathways of vital NET formation are still elusive. After chromatin release, the neutrophils continue chemotaxis, phagocytosis, and leukocyte recruitment. The DNA release within 1 h is faster than suicidal NET formation, which takes 1–6 h [16,49]. This process of direct DNA release was first described as the response to GM-CSF and C5a [13]. Another mechanism that can trap and eradicate Gram-positive bacteria such as Staphylococcus aureus without lytic death of neutrophils has been revealed by Pilsczek et al. This mechanism is also extremely fast (5–60 min) and NOX-independent. In this case of vital NET formation, vesicles are filled with nuclear DNA that are extruded to the extracellular space where they rupture and release chromatin [50]. In addition, neutrophils reportedly migrate and phagocytose in response to bacterial LPS. Toll-like receptors (TLRs) are reportedly also involved in vital NET formation, even though the mechanism of action is not completely understood. Bacterial LPS activates platelets via TLR4. The activated platelets bind viable neutrophils that consecutively release NETs [15,51,52].

How neutrophils stay alive and continue their active antimicrobial functions despite being enucleated is an essential question that needs to be answered. In light of previous reports, the changes in nuclear morphology may allow the formation of vesicles containing NET components. These may bud from the cytoplasmic membrane without disruption [15,16].

Another possible explanation for neutrophil survival is the reliance on mitochondria. The release of unfunctional, oxidized mitochondrial DNA (mtDNA) may support cellular survival. Considering the increased levels of mtDNA in the blood of many pathologies, the role of NET accompanying mtDNA may contribute to pathogeneses of certain diseases [13,53,54,55]. However, another notion that still needs to be clarified is how mtDNA passes through multi-membrane layers of mitochondria and viable cells. The differences in suicidal NET formation and the release of DNA from viable cells are highlighted in Figure 1.

## 3. Physiological Functions of NETs and Removal of NETs Remnants

Traditionally, neutrophils were considered as part of the only innate immunity. They are recruited very early to the site of infection/inflammation. Here, they immobilize and kill invaders to prevent their dissemination. When their mission is accomplished, neutrophils are eliminated from the “battlefield” to maintain tissue homeostasis. Otherwise, the prolonged presence of neutrophils in the inflammatory site may delay the healing process [47,56]. To maintain homeostasis, neutrophils often undergo apoptosis and are cleared by macrophages or non-professional tissue phagocytes. Some neutrophils reportedly return via vasculature to the bone marrow for clearance [57].

The antimicrobial arsenal of neutrophils includes degranulation, phagocytosis, oxidative burst, NET formation, and recruitment of further immune cells [58]. Neutrophils also influence the adaptive immune response, since they secrete a plethora of inflammatory mediators, including the B cell activation factors BAFF and APRIL [59], and they reportedly induce responses in surrounding lymphocytes and dendritic cells [60,61]. Not only neutrophils but also NETs participate in this cross-talk that occurs through direct cell contact or via soluble mediators [56,62]. Once NETs have been formed, extracellular DNA, citrullinated histones, and NET-borne enzymes activate further immune cells. On the downside, they may also foster chronic inflammation [63,64].

NETs have recently been shown to regulate the functions of additional neutrophils [65]. NETs stimulated IL-8 release and NET production in resting neutrophils [66], reflected by the modulation of their surface markers, the production of ROS, and the enhancement of exocytosis [65].

NETs not only affect other immune cells, they also serve as a physical barrier to prevent the spread of invaders or inflammatory mediators. At high neutrophil densities NETs tend to aggregate (form aggNETs), degrade soluble inflammatory mediators by NET-associated serine proteases, and, thus, support the resolution of inflammation [67,68].

A balance of NET formation and clearance is critical for tissue homeostasis. DNase1 and macrophages have long been known to clear NETs. NETs are dismantled by macrophages even though they possess a smaller volume than NETs. DNase1 cleaves chromatin into smaller fragments, allowing macrophages to engulf these NET remnants [7,69,70]. The main task of macrophages is to digest pathogens, and while M1 macrophages display a pro-inflammatory property, M2 macrophages encourage tissue repair by decreasing inflammation [71]. Following interaction with NETs, M2 macrophages secrete a number of chemotactic mediators, leading to the activation of M1 macrophages and monocytes. This facilitates the catabolism of the NETs. M2 macrophages engulf and digest the prey through an active and endocytosis-dependent mechanism with the help of LL-37, an antimicrobial peptide. In addition to DNase1, DNase1L3 is involved in the extracellular degradation of NETs [7,72,73].

## 4. Pathophysiology of NETs in Diseases

The massive recruitment of fully-armed neutrophils to the sites of infection/inflammation is responsible for first-line defense against invaders, but may also harbor the risk to drive certain pathologies. Indeed, the neutrophil-lymphocyte ratio (NLR) in the circulation has long been identified as a hematological measure of systemic inflammation [74]. The NLR is reliable and easily accessible in blood samples. It is frequently used as a diagnostic/prognostic marker for infection [75]. It reflects a link between innate (neutrophils) and adaptive immunity (lymphocytes) [74]. With the discovery of NETs, this link obtained a further player. In thrombi of renal cell carcinoma tumor and in stroke embolism, the NLR ratio correlated with the NET count [76,77].

Immunopathology of neutrophil-driven disorders can be caused by abnormal production or impaired clearance of NETs (Figure 2). Uncontrolled protease activity, hyperinflammation, and occlusions of vessels and ducts by aggNETs can lead to disease [78,79]. Uncontrolled ROS production drives the pathologies of chronic inflammatory diseases such as systemic lupus erythematosus (SLE). A decreased activity of DNase1 and non-function mutations of the DNASE1L3 gene were related to the development of kidney disease in patients with SLE and early onset SLE, respectively [80,81].

Impaired clearance of NETs and their remnants may also increase the number of modified agents that can serve as auto-immunogen and autoantigen. Components of NETs show autoantigenic potential, and high levels of NET-linked autoantigens have been reported in various autoimmune diseases. Among them, MMP8 [82] has been reported to be a NETs-associated autoantigen in rheumatoid arthritis (RA). MPO [83], PR3 [84], and properdin [85] also act as autoantigens in anti-neutrophil cytoplasmic antibodies (ANCA)-vasculitis (AAV). dsDNA [86], MMP9 [87], histones [88], and tissue factor (TF) [89] are reported to be autoantigens in SLE. Catalase [90], Annexin A1 [91,92], citrullinated histones [86,93], and citrullinated fibrinogen are autoantigens recognized by autoantibodies in RA and SLE. Although the mechanism is still elusive, NET-borne (modified) autoantigens seem to challenge the immunological tolerance and are prone to induce autoantibodies and autoimmune diseases, as mentioned before. In a self-amplifying loop, these autoantibody–autoantigen immune complexes can then again foster NET formation. This is referred to as the “vicious circle of inflammation and autoimmunity” [70,94].

The exaggerated inflammation in SLE has not only been linked to NET formation but also to a neutrophil subpopulation with low buoyant density, called low-density granulocytes (LDGs) [95,96]. LDGs exaggerate inflammation by type I IFNs (IFN-I), TNF-α, and IFN-γ. LDGs have, therefore, been proposed to cause onset and progression of tissue damage [95,97,98]. Functional analyses demonstrated that LDGs display increased NET formation [96]. LDGs also contribute to other diseases, such as AAV, and can thus be exploited as biomarker and target for developing new therapeutic interventions [99].

NETs also activate the complement system, an innate cascade of proteins that aids in the removal of pathogens and damaged cells from an organism. Complement also bridges innate and adaptive immunity and has a pivotal position in the pathologies of various diseases [100]. Components of the complement system including cell-bound complement activation products (CB-CAPs), C3, and C4 are involved in the pathogenesis of SLE [101,102]. C1q augments the activity of DNase1 in the serum, and cooperation of both molecules ameliorates an efficient NET chromatin degradation and clearance by phagocytes [103]. NETs of patients with AAV also interact with components of the complement system [85,104,105]. The serum of patients with AAV drives NET formation via factor B and properdin [85]; vice versa, anti-MPO driven NETs activate the complement system via interaction with C5a and C3d [104].

NETs and their components, such as histones, can also serve as damage-associated molecular patterns (DAMPs). NETs-derived extracellular DNA induces NLRP3 inflammasome in adult-onset Still’s disease (AOSD) [106] and high-mobility group box 1 (HMGB1), and LL37 can drive SLE [107,108]. However, there are controversial reports regarding the involvement of HMGB1 in AAV [107,109].

The detection of TF in NETs and neutrophil-derived microparticles obtained from patients with active AAV led to the assumption that TF drives disease-related coagulation [110]. It has also been suggested that NET-derived uncontrolled serine proteases degrade TF pathway inhibitors and, consequently, induce the coagulation cascade [111]. However, it is debated whether TF is directly expressed by neutrophils [112] or by endothelial cells [113], monocytes [114], or leukocytes [115] triggered by NETs. Regardless of its origin, NET-borne TF operates as a scaffold for platelet and erythrocyte adhesion, eventually resulting in clot formation. The latter recruits neutrophils that form further NETs and strengthen the primary clot [116,117].

Occlusive thrombosis occurs in coronary [118] and cerebral arteries [119] as well as in other vessels, especially of the capillary bed [120]. NET-related occlusions are not limited to the vessels but also appear in the ducts of pancreatic [121], Meibomian [122], and salivary glands. [123]. Most likely, other exocrine glands are similarly affected.

## 5. Approaches to the Regulation of NET Formation

NET-mediated vascular occlusion, persistent inflammation, and tissue damage highlight the importance of the delicate balance of NET formation and clearance. Inhibition of NET formation and aggregation is thus discussed as a possible therapeutic approach to prevent and ameliorate NET-driven inflammatory diseases. However, this may exacerbate episodes of infections, since NETs are endowed with microbicidal proteins, and aggNETs additionally possess anti-inflammatory activities. Metformin, chloroquine/hydroxychloroquine, or diphenyleneiodonium chloride (DPI) reportedly prevent NOX/ROS-dependent NET formation [124]. These drugs attenuated the severity of NET-mediated heparin-induced thrombocytopenia in mice [125]. Inhibition of MPO by PA-dPEG24 [126] or PF1355 [127] prevents NET formation in the early stage. Blocking of PADI4 reportedly improved the progression of various NET-driven diseases [128], but its inhibition also bears the risk of exacerbated infections.

DNAses degrade NETs and prevent their accumulation. Administration of recombinant human DNAse (rhDNAse) improved the progression of cystic fibrosis without side effects [129,130,131]. Pneumococcal meningitis and trauma were also successfully treated by the application of rhDNase [132,133]. In mouse models, the application of DNase as a treatment for breast cancer was successful [134], but showed controversial outcomes for ischemia-reperfusion [135] and murine lupus [136]. In patients with SLE, administration of rhDNase did not improve disease progression, although DNase1 contributes to the degradation of accumulated NETs in the course of this disease [80]. This may be improved by the concomitant addition of agents that support resolution of NET-associated histones and proteases [132,133].

## 6. Conclusions and Future Directions

Neutrophils and NETs play an important role in the defense against invading pathogens, as well as in the resolution of inflammation. NETs and NET-associated proteases trap and kill pathogens; aggNETs display anti-inflammatory potential by scavenging and degrading inflammatory cytokines. However, disruption of the delicate balance between NET formation, degradation, and clearance can contribute to the development and pathogenesis of various inflammatory diseases. NETs and their components interact with other cells of the innate and adaptive immune system and, thus, modulate the immune responses [7,64,137]. This cross-talk with other components of the immune system underscores the importance of exploring the role of NETs and their components in physiological and pathological processes. It is crucial to gain further insight into epigenetic modifications that take place both during NET formation and during the course of their immunological functions. An understanding of their intricate role in immunology and pathology can only be achieved through comprehensive investigations using big data, in vivo, and in vitro analyses. The broad “omics” approach enables the identification of all agents with which NETs interact, allowing more reliable and conclusive statements regarding the potential roles of NETs in various immunological processes. In this way, NETs can contribute to a better understanding of the development and pathophysiology of a plethora of inflammatory diseases, which is essential for the design of treatment approaches. [70,96,138].

## Figures and Tables

**Figure 1 ijms-23-12855-f001:**
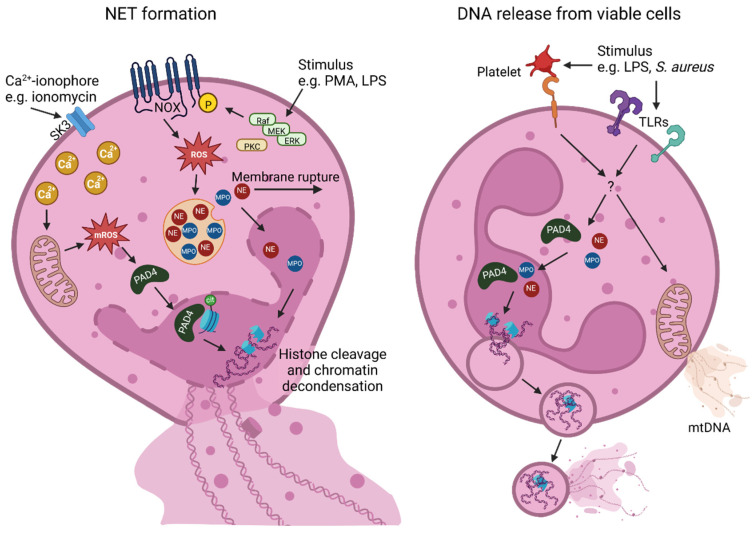
NET formation mechanisms and their components. Two main types of NET formation are reported: NOX-independent and NOX-dependent. Both types lead to histone cleavage and chromatin decondensation and are lethal to the cell. The NOX-independent pathway is triggered by calcium ionophores such as ionomycin, which promotes calcium influx and mitochondrial ROS production. This induces translocation of PAD4 to the nucleus, where it citrullinates histones. The NOX-dependent pathway, triggered by stimuli such as PMA and LPS, activates the Raf–Merk–Erk pathway and leads to ROS-dependent release of MPO and NE from the azurosome. These enzymes mediate rupture of the membrane, nuclear envelope breakdown, and chromatin decondensation inside the nucleus. While calcium, mROS, and ROS can be clearly assigned to the NOX-independent or -dependent pathway, respectively, it is not yet clear whether PAD4, MPO, and NE act in both pathways. Another form of NET generation involves DNA release from viable cells after neutrophil activation by stimuli such as S. aureus, LPS, or LPS-activated platelets. The exact mechanisms remain elusive. There are two ways the cell can survive DNA release. Either the DNA is released via vesicles, which maintains membrane integrity and allows the cell to retain some physiological functions, or the cells release only non-functional mitochondrial DNA. Created with BioRender.com.

**Figure 2 ijms-23-12855-f002:**
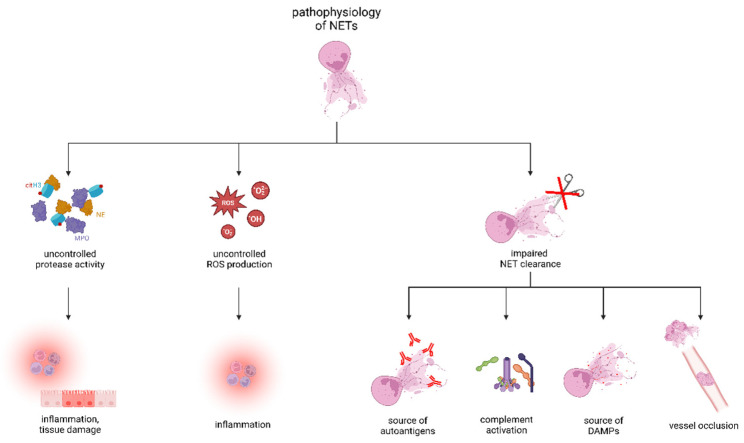
Pathophysiology of NETs in diseases. Abnormal or impaired clearance of NETs can cause several neutrophil-driven disorders. Enhanced uncontrolled release of NET-associated proteases and ROS promotes inflammation and can lead to tissue damage. Impaired NET clearance and, thus, the persistent presence of NET chromatin and associated proteins in the extracellular space serve as a source of autoantigens and DAMPs, activate the complement, and, in the case of aggregation, occlude vessels. Created with BioRender.com.

## Data Availability

Not applicable.

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
