# Peer review of "Neutrophils’ Extracellular Trap Mechanisms: From Physiology to Pathology"

_ijms, 2022, doi:10.3390/ijms232112855_

Round 1

Reviewer 1 Report

This review entitled, “Neutrophils Extracellular Trap Mechanisms: from physiology to pathology,” is an appropriate and remains a timely review as the importance of NETs as players of the innate adaptive crosstalk.  NETS can promote resolution of inflammation or mediate pathologic disease through serving as sources of autoantigens and/or DAMPs and lead to aggregation and vascular/ductal occlusions. The short review is outlined in a straight-forward manner with a nice figure providing an overview. There was no bias detected.

Reviewer 2 Report

This consise review describes the formation, composition, and basic roles of neutrophil extracellular traps (NETs) in both healthy and disease states. The authors stated their aim was to provide an outline of the aforementioned aspects of NETs, which I believe that they have achieved. The manuscript is well written and provides a good level of detail in most areas. The structure is logical and the subheadings are relevant and appropriate.

General comments:
There are several published reviews on NETs, including some published in the last 18 months. I felt that the authors could emphasize more how their review offers something different and who would benefit most from reading it. For example, compared to some others (Hidalgo 2021, Chen 2021 etc.), this is a concise general overview of NETs and would be useful for readers that are relatively new to the field. 
Referencing was quite thorough, included recent articles, and there was no in-appropriate self citations. 
I believe that a second figure showing the NET formation processes would be a valuable addition.

Overall, the review was quite statement/fact driven, like a collection of other's results, without much interpretation or tying together of the ideas by the authors. I would like to see more in section 6 regarding where the authors think the field is heading or where research efforts should focus.  

Specific comments:
Line 72, first sentence in 2.1 doesn't make sense. 
Line 99, please define what is meant by "form NETs sufficiently"

Line 154, should this be "The optimum pH of PADI4"? 

Paragraph in lines 207-215, I would like to see a short explanation of M1/M2 macrophages.
